# Optimizing time and split application of nitrogen fertilizer to harness grain yield and quality of bread wheat (*Triticum Aestivum L*.) in northwestern Ethiopia

**Bitwoded Derebe[1], Yayeh Bitew[2]\*, Fikeremariam Asargew[3], Gobezie Chakelie[4]**

**1** Amhara Regional Agricultural Research Institute, Adet Agricultural Research Center, Bahir Dar, Ethiopia, **2** College of Agriculture and Environmental Science, Bahir Dar University, Bahir Dar, Ethiopia, **3** Amhara Regional Agricultural Research Institute, Bahir Dar, Ethiopia, **4** Amhara Regional Agricultural Research Institute, Gonder Agricultural Research Center, Gonder, Ethiopia

\* yayehbitew@gmail.com

**Data Availability Statement:** All relevant data are within the paper and its Supporting Information files.

## Abstract

Improper nitrogen application time during the crop growing period is one of the most limiting factor for wheat production. A field experiment was conducted in Northwestern Ethiopia with the objective of determining the appropriate N fertilizer application time for improving bread wheat production. Twelve treatments (½ urea at 50% emergence + ½ urea at tillering, ½ urea at tillering + ½ urea at booting, 1/3rd urea at 50% emergence + 1/3rd urea at tillering + 1/3rd urea at booting, 2/3rd urea at tillering + 1/3rd urea at booting, all urea at tillering, all urea at booting, all N at tillering, all N at booting, ½ N at sowing+ ½ N at tillering, ½ N 50% emergence + ½ N at tillering, 1/3rd N at 50% emergence + 1/3rd N at tillering + 1/3rd N at booting, ½ N at tillering + ½ N at booting) were lied out in randomized complete block design (RCBD) with three replications. The study showed that wheat grain yield and protein content was highly influenced by the environment and indirectly correlated with each other as affected by N time of applications. The grain yield at Adet, Wonberema and Debre Elies was increased by 31%, 14% and 18%, respectively when N was applied with DAP at sowing over the blanket recommendation. At all locations, grain protein content decreased as the number of N split application increased 1 to 3 times. Thus, depending on the purpose of the producers, it can be concluded that application of ½ urea at 50% emergence + ½ urea at tillering with the application of DAP at sowing gave maximum wheat grain yield, while optimum grain protein content was obtained when N was applied after the crop is emerged and would be used in most dominant wheat producing areas of northwestern Ethiopia. Further study should be conducted on split application of blended fertilizers (NPS, NPSBZN etc.).

## Introduction

According to FAO [1] the importance of wheat crop in the world trade is greater than that of all other crops produced and has been a food of the major civilizations for 8,000 years. It is also

**Funding:** The author(s) received no specific funding for this work.

**Competing interests:** The authors have declared that no competing interests exist.

the most cultivated cereal crop in the world and the quantity produced is more than that of any other crop, feeding about 40% of world population. It is becoming the most important cereal crop grown on a large scale in the tropical and subtropical regions of the world [2].

Ethiopia is the second largest producer of wheat in sub-Saharan Africa following South Africa and more than 1.69 million hectare of land was cultivated for the production of wheat under rain fed conditions with the production and productivity of 4.6 million ton and 2.74 t ha$^{-1}$, respectively [3]. It is wider importance in the order of mid altitudes and highland of Ethiopian mainly in Oromia, Amhara, and southern regions of Ethiopia [4]. Among food grain crops, bread wheat covers about 13.33% of the national grain crop area and 15.81% of the national grain production in the country [5]. It provides more protein than any other cereal crop [6]. Despite its importance, the productivity of wheat in Ethiopia under rain fed season is low particularly in Amhara region (2.79 t ha$^{-1}$) which is 6.2% lower than the national average [7], 20.34% far below the world's average productivity of 3.5 t ha$^{-1}$ [8] and 223% lower than potential yield [9]. Consequently, the country continually remains a net importer of about 1.7 million tons of wheat, draining the national treasury [10] as a result of the huge gap between production and consumption levels [11].

Low soil fertility, improper fertilizer application, lack of appropriate seeding rate, fertilizer rate, planting methods and pests are among the major constraints limiting wheat production in Ethiopia [12, 13]. Nitrogen and phosphors are the major limiting soil nutrients in all parts of Ethiopia [14]. They are also the dominant inorganic fertilizers applied to boost the productivity of crops. Nitrogen is a major responsible factor in achieving optimum grain yield and quality [15] and surprisingly, compared to other fertilizer types it needs special management practices [16]. Depending on the type of fertilizer used, nitrogen is characterized by rapid uptake by plants, highly mobile in the soil (subject to leaching and volatilize as $NH_3$), denitrification, immobilization by microbes, forming complex with soil organic matter (humus) and fixed in 2:1 clay minerals [17, 18]. Thus, Ghafoor et al. [19] reported that nitrogen is the most frequently deficient nutrient than others. Maximizing crop yield requires understanding of nitrogen behavior. Increasing the availability and efficiency of N uptake is considered to be a primary concern in increasing wheat grain yield and protein content. However, a challenge for wheat production is to increase grain yield while maintaining its protein content. Rossi et al. [18] pointed out that due to the dilution effect there have been frequent negative correlations between wheat grain protein content and yield. Moreover, estimation of different research findings indicated that only 33% of total N applied for cereal production in the world removed in the grain and 67% is lost by various reasons [17, 20]. Consequently, the farmers are compelled to apply more than the actual need of the crop to compensate the losses [17]. The loss of N not only troubles the farmer, but it has also hazardous impacts on the environment. To reduce these challenges increasing the N use efficiency through different strategies is indispensable. Some of the strategies includes (i) use of appropriate type of fertilizer and application time during the crop growing period, (ii) application of optimum N fertilizer rate, (iii) appropriate method of application,(iv) use of appropriate cultivar and (v) application at the appropriate climatic conditions [21].

On both sandy soils subject to leaching and poorly drained soils prone to denitrification, split applications may be a strategy to consider [16, 20]. Thus, nitrogen scheduling plays a vital role in the growth, production, and quality of wheat as well as in its use efficiency [20]. One of the main causes of low nitrogen use efficiency in actual N management practices is the scarce synchrony between N soil input and crop demand [16]. The application of nitrogen fertilizer to wheat needs to be at critical N uptake by the plant. Applications after critical plant N uptake may restrict plant N utilization, growth, and yield and before critical uptake may result in N losses from the soil [22]. Starting from the onset of scientific research, inconsistent

information has been released on N application time for bread wheat production in Ethiopia. According to Asnakew et al. [23] application of 50%of the total N dose at sowing and the rest at full tillering stage significantly increased grain yield as well as protein content of wheat. Tilahun *et al.* [24] indicated that split application of nitrogen at planting and tillering had shown significant improvement in wheat yield. Haile et al. [25] reported that higher grain yield, N utilization efficiency and protein were recorded when N was applied 1/4th at planting, 1/2nd at mid-tillering, and 1/4th at the anthesis stage of the crop. Rahman et al. [26] reported that split application of N was effective in increasing wheat grain yield and especially grain protein was improved by the late application of N. Multi-location N fertilizer trials conducted by Yohannes and Nigussie [27] in the highlands of southeastern Ethiopia indicated that split application of N: 1/3 at planting and the rest 2/3 at mid-tillering, provided optimum wheat yield. In the other study, applied N at thirty days after sowing (tillering) had contributed better grain yield than applied N at planting and fifteen days after planting [13].

Currently, Ethiopian Bureau of Agriculture and Agricultural transformation agency promoting the application of urea at planting which is wastage as it might leach out before root emergence [28, 29]. Many farmers often use uniform rates of N fertilizers in two times based on expected yields that could be inconsistent from field to-field and year-to-year depending on factors that are difficult to predict prior to fertilizer application. Also, farmers often apply fertilizer N in doses much higher than the blanket recommendations to ensure high crop yields. On the other side, some scientific findings disclosed that urea should be applied basal so long as most of the yield components of bread wheat are determined at early growth stages of the crop and even root is emerged within 3–7 days from planting [30]. Moreover, many farmers in the wheat production potential areas of Amhara region have observed good responses when N has been applied at planting and tillering stage. By considering the current information gap on N application time this research was initiated to validate different N application time on different wheat growing stages with various ratios of N across a range of wheat growing environments in Northwestern Ethiopia. Thus, the main objective of this research was to determine the appropriate N fertilizer application time for maximum bread wheat yield production with optimum protein content.

## Materials and methods

### Description of experimental sites

A field experiment was conducted in three wheat production potential areas of Northwestern Ethiopia; namely *Adet*, *Debre Elias*, *and Wonberema* districts of Amhara Region, Ethiopia during 2014 and 2015 main cropping seasons. The experiment was tested at four sites per experimental location in each cropping seasons with a total of twelve sites per year. Generally, the rainfall in the study areas follows a dominantly unimodal distribution with the main peak in June to September, during which more than 80% of the annual rainfall is received. Smaller peaks occur in May and October [9]. These experimental areas are generally found in agro-ecological zone of moist *Wayena Dega* (mid land).The farming system of the study areas is characterized by 100% mixed crop-livestock systems [31]. These three districts are among the leading potential areas for bread wheat production in the country [32]. Geographically, Adet is located at 11°16' N latitude and 37°29' E longitude with an altitude of 2240 meters above sea level (m.a.s.l.) It has a long year mean (30 years) annual rainfall of 1211mm with a minimum and maximum temperature of 11.57°C and 26.89°C, respectively [33]. Wonberema is located at 10°27'N latitude and 37°56' E, longitude with an altitude of 2600 m.a.s.l. It has a long year mean annual rainfall of 1211mm with a minimum and maximum temperature of 17°C and 25°C, respectively [34]. Debre Elias is located at 10° 33' N latitude and 37°72'E longitude with

an altitude of 1941 m.a.s.l. It has a mean annual rainfall of 1211mm with a minimum and maximum temperature of 17˚C and 25˚C, respectively [34]. Weather data (rainfall and temperature) of the experimental year were collected in the northwestern Ethiopia meteorology station office at Bahir Dar (Fig 1). On average rain fall at Debre Elias was 137% and 150% higher over Wonberema and Adet, respectively. In Debre Elias and Wonberema, the rain fall increased from 2014 to 2015 by 71.43% and 14.29%, respectively, while at Adet it was relatively constant across years. At Adet the soil textural class is clay [35] while at Wonberema [36] and Debre Elias [37] the soil textural class is sandy loam. The experimental site at Adet constitutes an average pH of 5.4, organic carbon of 2.47%, total nitrogen of 0.8%, available phosphorus of 1.98 ppm and cation exchange capacity of 31.2 Cmole(+) kg$^{-1}$ [35] while the respective soil nutrients at Debre Elias were 5.12, 2.63%, 0.14%, 2.53 ppm and 25.02 Cmole(+) kg$^{-1}$ [37]. At Wonberema, pH, organic carbon, total nitrogen and available phosphorus were 5.72, 3.1%, 0.3%, 3.69 ppm, respectively [36].

### Treatments, experiment design and planting procedures

Treatments were twelve nitrogen (N) application times fixed based on [38] with different N split ratios described in Table 1 bellow:

Note that, Di-Ammonium Phosphate (DAP) and TSP (Tri- Supper Phosphate) were the sources of P2O5 and DAP and UREA were the sources of N. However, at sowing $P_2O_5$ was applied in the form of DAP from treatment 1 to 6, while from treatment 7 to 12, $P_2O_5$ was applied in the form of TSP. Thus, N was applied at sowing only in the former treatments (1 to 6) (Table 1). Nitrogen was applied based on the time of application described in Table 1. The treatments were lied out in randomized complete block design with three replications.The distance between adjacent blocks and plots within a block were 1m and 0.5m apart, respectively.

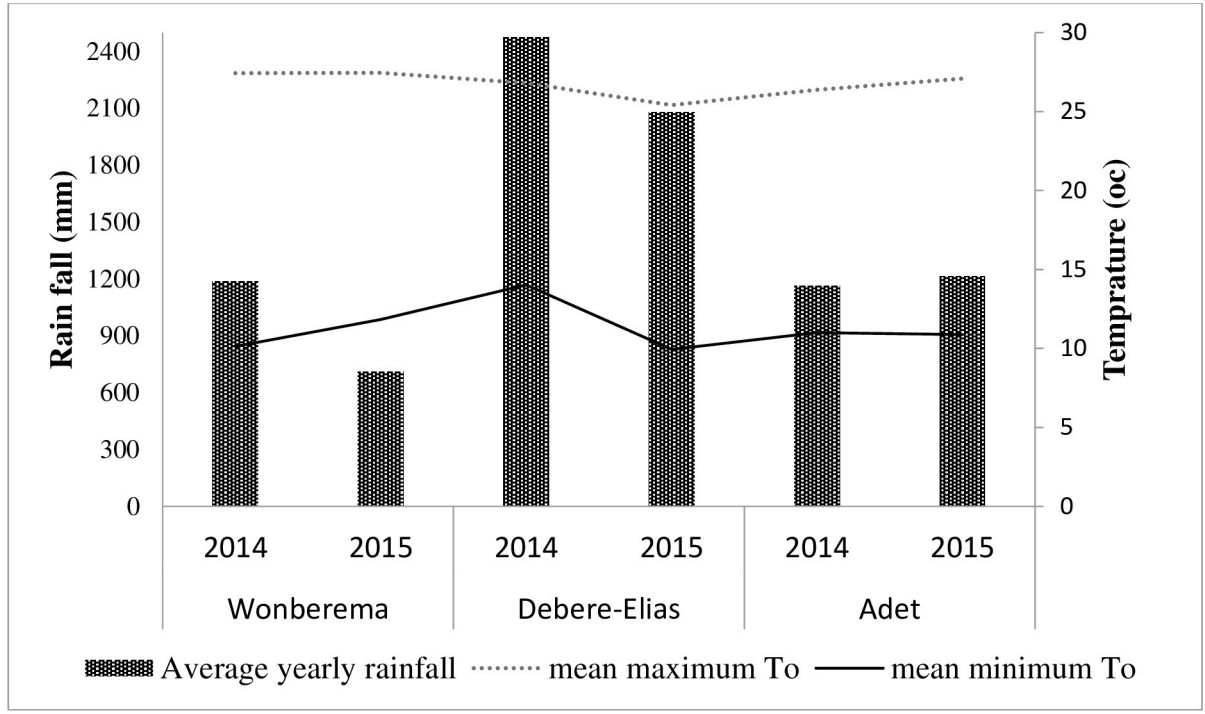

**Fig 1. Mean annual rainfall and maximum and minimum temperatures during 2014 and 2015 main growing seasons in the study area.**

**Table 1. Description of the experimental treatments.**

| TrtNo. | Nitrogen application | | |
| --- | --- | --- | --- |
| | 1st application | 2nd application | 3rd application |
| 1 | 1/3rd urea at 50% emergence | ½ urea at tillering | - |
| 2 | ½ urea at tillering | ½ urea at booting | - |
| 3 | 1/3rd urea at 50% emergence | 1/3rd urea at tillering | 1/3rd urea at booting |
| 4 | 2/3rd urea at tillering | 1/3rd urea at booting | - |
| 5 | All urea at tillering | - | - |
| 6 | All urea at booting | - | - |
| 7 | All N at tillering | - | - |
| 8 | All N at booting | - | - |
| 9 | ½ N at sowing | ½ N at tillering | - |
| 10 | ½ N 50% emergence | ½ N at tillering | - |
| 11 | 1/3rd N at 50% | 1/3rd N at tillering | 1/3rd N at booting |
| 12 | ½ N at tillering | ½ N at booting | - |

TrtNo.−Treatment number,N−nitrogen. Sowing−Zadoks Growth Stage 00; emergence−Zadoks Growth Stage 11−18; tillering−Zadoks Growth Stage 21−26; booting −Zadoks Growth stage 41−45.

The gross and net plot size were 1.6m (length) x 4m (width) and 1.2m (length) x 4m (width), respectively. Seed bed preparation (ploughing) was made four times using oxen following the local recommended cultural practices for wheat production. Wheat seeds were weighed separately for each plot based on the recommend seed rate of 150 kg ha$^{-1}$. In each plot, wheat was sown using a drill planting method with inter- row spacing of 20cm. At all experimental locations, the recommended fertilizer dose for rain-fed wheat of 276 kg ha$^{-1}$ N and 92 kg ha$^{-1}$ P$_2$O$_5$ were applied at all locations. All P$_2$O$_5$ was applied at planting, while N was applied based on the respective treatment levels descried in Table 1. Band fertilizer application method was used at any growth stage of wheat at a depth of 2cm. One combined inter-culture (hoeing) and hand weeding at 3 weeks after sowing were carried out to keep the plots free from weeds and to provide better aeration. The next two hand weeding activities were operated at the mid-tillering and booting stage.

## Data collection

Average plant height was measured from ten randomly taken plants of the net plot at 90% physiological maturity. Number of total tillers per plot (TNT) was counted from one meter square quadrant in each net plot. The total above ground-biomass from the net plot area was harvested and sun-dried for two weeks with an average air temperature of 25–27˚C till complete drying was attained. Then grain yield was separated and determined from the total biomass yield. The grain yield was then dried, threshed, cleaned and adjusted to the 12.5% moisture level. The moisture correction factors was done by using the following formula [39]:

$$Adjusted\ Yield = \frac{100 - Measured\ moisture\ content\ \%}{100 - 12.5\%} \tag{1}$$

Thus, 12.5% moisture adjusted grain yield = moisture correction factors X grain yield obtained from each plot. Thousand-kernel weight (TKW gram) was determined by counting 1000 grains by using an electronic automatic seed counter and weighing those counted kernels sampled from the net plot using a sensitive balance of precision + 0.001g. The thousand kernels weight was recorded after the grain was adjusted to 12.5% moisture content by using

Draminski Gmm mini quick moisture tester instrument. Hectoliter weight (HLW) or test weight (TW), a physical quality parameter commonly used in the cereals, is an estimate of bulk density (g cm$^{-3}$) and the most used indicator by the milling industry. Its determination was done by device equipped with a funnel which provides uniform packing in 1100-ml (USA) measuring cups. It is the weight of 100 litters of wheat and is the simplest criteria of wheat quality. It gives us a rough index of flour yield. Higher the hectoliter weight, the better is the flour yield. The values large from 70–85 kg hl$^{-1}$ higher the hectoliter weight better is the wheat for milling in terms of flour yield [40]. From the above dried, cleaned and adjusted yield a subsample from each net plot was taken and milled. Then, grain N content of the samples were determined using the micro-Kjeldahl method as stated by American Association of Cereal Chemists (AACC) [41]. The protein content of wheat flour samples for each treatment was performed by following AACC [42] using a 5.7 conversion factor as follows and expressed in dry matter %.

$$\% \ of \ Protein \ contenet = \% \ of \ Nx5.7 \qquad (2)$$

## Statistical analysis

Data analysis for all agronomic attributes and quality parameters of wheat was conducted using the general linear model procedure of SAS version 9.2 [43] for each site and year. The data were combined over sites since values for the error mean square of the two years were homogeneous [44]. In the combined analysis, the mixed procedure was used and replication and year were considered as a random variables and site and treatments as fixed variables. Means were separated by Duncan Multiple Range Test (DMRT) when crop characteristics showed significant differences at the probability level of 0.05. Using the same software, pair-wise correlation analysis was also done to assess the association between wheat grain yield and other agronomic attributes and quality parameters. The values of correlation coefficient lies within the range of -1 and +1, with a value close to +1 or-1 represents a strong linear relationship, a value of r close to zero means the linear relationship is very weak and an intermediate value of r indicates the portion of variation in one variable that accounted for by the linear function of the other variable [44].

## Results and discussion

### Effect of locations and growing seasons

Almost all agronomic attributes and quality parameters of wheat were significantly influenced (*P*<0.01) by growing seasons (Tables 2–5) indicating the two cropping seasons varied in amount and distribution of rainfall and minimum and maximum temperatures (Fig 2). Moreover, almost all agronomic attributes and quality parameters of wheat were significantly affected (*P*<0.01) by locations indicating the locations varied in edaphic as described in section 2.1 and climatic conditions (Fig 2).This implied that N use efficiency across wheat growing stages was highly influenced by the environment and consistent with the finding of Lopez-Bellido *et al.* [21]. All measured agro-morphological traits of bread wheat were significantly influenced by nitrogen fertilizer split and time of application by location by year interaction. The selection of appropriate nitrogen fertilizer split (time of application) for better grain yield and quality in one environment may not enable the identification of nitrogen time of application that can repeat nearly the same response in other environments.

**Table 2. Plant height and tiller number of bread wheat as affected by nitrogen split and time of application[a].**

| N time of application | Adet | | Wonberema | | Debere Elias | |
|---|---|---|---|---|---|---|
| | **PH** | **TT** | **PH** | **TT** | **PH** | **TT** |
| **N1** | 97.40[ab] | 432.50[a] | 101.93ab | 358.83[ab] | 104.70[a] | 431.64[a] |
| **N2** | 93.07[abcde] | 374.50[abc] | 98.90abc | 348.17[ab] | 99.40[abc] | 375.17[bcd] |
| **N3** | 99.57[a] | 376.17[abc] | 100.67[abc] | 329.00[ab] | 104.83[a] | 389.08[abcd] |
| **N4** | 95.68[ab] | 394.67[ab] | 100.53[abc] | 340.08[ab] | 100.93[abc] | 367.50[bcd] |
| **N5** | 95.07[abc] | 369.00[abc] | 100.00[abc] | 358.58[ab] | 103.87[ab] | 372.67[bcd] |
| **N6** | 87.67[cde] | 310.83[c] | 90.33d | 313.58[b] | 97.80[cde] | 344.67[d] |
| **N7** | 87.03[ed] | 365.50[abc] | 92.90[cd] | 376.92[a] | 92.80[ef] | 386.08[abcd] |
| **N8** | 79.87[f] | 324.00[bc] | 82.40[e] | 257.08[c] | 89.33[f] | 292.50[e] |
| **N9** | 94.03[abcd] | 358.00[abc] | 106.83[a] | 367.92[ab] | 102.43[abc] | 395.42[abcd] |
| **N10** | 90.90[bcde] | 362.50[abc] | 101.83[ab] | 350.17[ab] | 99.60[abc] | 406.67[ab] |
| **N11** | 95.50[ab] | 390.00[ab] | 100.63[abc] | 344.50[ab] | 98.30[cde] | 403.50[abc] |
| **N12** | 85.63[ef] | 362.50[abc] | 93.80[abc] | 349.50[ab] | 93.23[def] | 353.75[cd] |
| **Significance level** | ** | ** | ** | ** | ** | ** |
| **CV (%)** | 4.80 | 11.63 | 4.74 | 10.28 | 3.32 | 7.86 |
| **Year** | ** | ** | ** | ** | Ns | ** |
| **Year*Treatment** | * | * | Ns | Ns | ** | Ns |

[a]Data were combined over years and sites in each experimental locations.N1 = ½ urea at 50% emergence + ½ urea at tillering, N2 = ½ urea at tillering + ½ urea at booting, N3 = 1/3[rd] urea at 50% emergence + 1/3[rd] urea at tillering + 1/3[rd] urea at booting, N4 = 2/3[rd] urea at tillering + 1/3rd urea at booting, N5 = All urea at tillering, N6 = All urea at booting, N7 = All N at tillering, N8 = All N at booting, N9 = ½ N at planting + ½ N at tillering, N10 = ½ N 50% emergence + ½ N at tillering, N11 = 1/3rd N at 50% emergence + 1/3[rd] N at tillering + 1/3[rd] N at booting, N12 = ½ N at tillering + ½ N at booting. PH, plant height (cm); TT, tiller m[-2]. Treatments within a column sharing the same letter are not significantly different.

*significant at the .05 probability level;

**significant at the .01 probability level; ns, non significant at 0.05 probability level.

## Yield and yield component of bread wheat

Results shown in all locations N split and application time was significantly ($P<0.01$) affected plant height, total tillers m[-2], biomass and grain yield (Tables 2 and 3). There was no definite trend for N split and time of application effect on plant height in all locations. At Adet and Debre Elies the tallest plant height was obtained on treatments receiving N with DAP at sowing + 1/3[rd] urea at 50% emergence + 1/3[rd] urea at tillering + 1/3[rd] urea at booting (Table 2). Similarly, Wazir and Akmal [45] reported that the highest days to anthsis and maturity, and plant height were observed when 25%, 50% and 25% of N was applied at sowing, tillering and anthsis of wheat, respectively. These findings are also highly in conformity with the findings of Singh et al. [46] who claimed that the tallest and the shortest plant height were obtained when N was applied in three splits with N applied at planting and at one or two growing stages of wheat, respectively. While at Wonberema application of ½ N at planting + ½ N at tillering stage of the crop produced the tallest plant height (Table 2). Generally, significantly maximum plant height was recorded in the application of N at early and tillering stage of the crop than at a later stage of the crop in all locations. The possible explanation of the positive effect of split N application compared to single N application at later growth stages is gaseous losses from applied topdressing. The shortest plant heights were scored on treatments received all N at the booting stage of the crop in all locations (Table 2).

At Adet, the total tiller number m[-2] is ranged from 324 for applying all N at booting to 432 in the application of half urea at 50% emergence + half urea at tillering including N in DAP

**Table 3. Grain and above biomass yield of bread wheat as affected by nitrogen split and time of application[a].**

| N time of application | Adet | | Wonberema | | Debere Elias | |
|---|---|---|---|---|---|---|
| | GY | BY | GY | BY | GY | BY |
| N1 | 4.89[a] | 11.84[a] | 4.37[a] | 10.17[a] | 5.23[a] | 12.95[a] |
| N2 | 4.28[abc] | 10.14[abc] | 4.35[a] | 9.92[ab] | 4.26[bcd] | 10.50[bc] |
| N3 | 4.35[ab] | 11.34[a] | 4.33[a] | 9.70[ab] | 5.01[ab] | 12.81[a] |
| N4 | 4.38[ab] | 10.76[ab] | 4.04[a] | 9.39[ab] | 4.81[abc] | 11.43[ab] |
| N5 | 4.01[abc] | 10.34[abc] | 4.17[a] | 9.84[ab] | 5.07[ab] | 12.30[a] |
| N6 | 2.50[de] | 7.02[d] | 3.15[c] | 7.22[c] | 4.12[cd] | 9.51[c] |
| N7 | 3.23[cd] | 7.71[dc] | 3.66[b] | 8.56[bc] | 3.80[d] | 9.33[c] |
| N8 | 2.19[e] | 6.63[d] | 2.70[d] | 5.54[d] | 2.78[e] | 6.86[d] |
| N9 | 4.03[abc] | 9.97[abc] | 4.63[a] | 10.93[a] | 4.76[abc] | 11.68[ab] |
| N10 | 3.93[abc] | 9.74[abc] | 4.12[ab] | 10.16[ab] | 5.05[ab] | 12.49[a] |
| N11 | 3.98[abc] | 9.98[abc] | 4.17[ab] | 9.84[ab] | 4.76[abc] | 11.51[ab] |
| N12 | 3.62[bc] | 8.37[bcd] | 3.68[bc] | 8.50[bc] | 3.77[d] | 9.10[c] |
| Significance level | ** | ** | ** | ** | ** | ** |
| CV (%) | 15.92 | 16.09 | 8.78 | 10.58 | 11.32 | 9.72 |
| Year | ** | ** | ** | ** | ** | * |
| Year*Treatment | * | Ns | ** | Ns | Ns | Ns |

[a]Data were combined over years and sites in each experimental locations. N1 = ½ urea at 50% emergence + ½ urea at tillering, N2 = ½ urea at tillering + ½ urea at booting, N3 = 1/3[rd] urea at 50% emergence + 1/3[rd] urea at tillering + 1/3[rd] urea at booting, N4 = 2/3[rd] urea at tillering + 1/3rd urea at booting, N5 = All urea at tillering, N6 = All urea at booting, N7 = All N at tillering, N8 = All N at booting, N9 = ½ N at planting + ½ N at tillering, N10 = ½ N 50% emergence + ½ N at tillering, N11 = 1/3rd N at 50% emergence + 1/3[rd] N at tillering + 1/3[rd] N at booting, N12 = ½ N at tillering + ½ N at booting. GY, grain yield (t ha[-1]); BY, biomass yield (t ha[-1]). Treatments within a column sharing the same letter are not significantly different.

*significant at the .05 probability level;

**significant at the .01 probability level; ns, non significant at 0.05 probability level.

during sowing (Table 2). At Wonberema, the total tiller number m[-2] is ranged from 257.08 in the application of all N at the booting stage to 376.92 in the application of all N at tillering stage. At Debre Elias, the total tiller number m[-2] is ranged from 292.50 in the application of all N at the booting stage to 431.6 in the application of half urea at 50% emergence + half urea at tillering stage of the crop (Table 2). This clearly showed that the application of N in splitting with various ratios was better in total tiller number m[-2] than a single time application of N. Specifically, the highest tiller m[-2] at Adet and Debre Elies was obtained from the application of ½ urea at 50% emergence + ½ urea at tillering plus addition of N with DAP at planting, while at Wonberema it was observed when all N was applied at tillering stage. Similar to the former result, Singh et al. [46] claimed that No. of tiller, No. of bearing tiller and plant population (bearing/effective tiller/ seqmt) were highest when 75 kg N + 60 kg $P_2O_5$ + 40 kg $K_2O$ was applied at sowing + 40 kg N at crown root initiation + 35 kg N at panicle initiation. The lowest tiller m[-2] at Wonberema and Debre Elies was recorded when all N was applied at booting stage while at Adet it was obtained when N was applied at planting with DAP and all urea was applied at booting stage (Table 2). Nitrogen deficiency during this early part of the growing season would be most likely to interrupt the tillering process and lead to death of the younger tillers [47].The application of N at the emergence and tillering stage may contribute to the increase in leaf area index, which results in greater photosynthetic area. The greater photosynthetic area contributes in increasing the availability of carbohydrates to maintain production of tillers and ears that result in grains [18], which explains the effect of N management at Adet and Debre Elies environments. This supports the necessity of N application in the

**Table 4. Thousand seed weight (TSW) and hectoliter weight (HLW) of bread wheat as affected by split and time of nitrogen application[a].**

| N application time | Adet | | Wonberema | | Debere Elias | |
|---|---|---|---|---|---|---|
| | TSW | HLW | TSW | HLW | TSW | HLW |
| N1 | 32.61 | 72.30[abc] | 31.31 | 72.67[bcd] | 31.51 | 76.27 |
| N2 | 32.09 | 71.13[abc] | 29.36 | 71.90[cd] | 31.73 | 73.67 |
| N3 | 33.86 | 70.20[bc] | 30.18 | 73.13[abcd] | 32.65 | 74.87 |
| N4 | 33.44 | 71.80[abc] | 30.36 | 71.07[d] | 33.73 | 75.10 |
| N5 | 33.10 | 70.73[abc] | 32.16 | 71.40[d] | 30.72 | 74.73 |
| N6 | 33.84 | 71.60[abc] | 30.11 | 74.53[ab] | 32.62 | 75.33 |
| N7 | 33.32 | 69.73[c] | 29.77 | 71.27[d] | 32.62 | 73.80 |
| N8 | 31.63 | 73.63[abc] | 31.11 | 74.23[abc] | 33.42 | 73.33 |
| N9 | 34.56 | 70.80[abc] | 30.58 | 75.57[a] | 31.75 | 74.58 |
| N10 | 33.67 | 74.23[a] | 32.45 | 74.23[abc] | 32.83 | 73.47 |
| N11 | 32.61 | 72.73[abc] | 29.59 | 72.67[bcd] | 32.80 | 74.27 |
| N12 | 34.58 | 71.50[abc] | 32.56 | 72.20[bcd] | 33.75 | 73.05 |
| Significance level | Ns | ** | Ns | ** | Ns | Ns |
| CV (%) | 5.40 | 2.75 | 7.20 | 2.00 | 8.55 | 2.71 |
| Year | ** | ** | ** | ** | Ns | ** |
| Year*Treatment | Ns | Ns | ** | ** | Ns | Ns |

[a]Data were combined over years and sites in each experimental locations. N1 = ½ urea at 50% emergence + ½ urea at tillering, N2 = ½ urea at tillering + ½ urea at booting, N3 = 1/3rd urea at 50% emergence + 1/3rd urea at tillering + 1/3rd urea at booting, N4 = 2/3rd urea at tillering + 1/3rd urea at booting, N5 = All urea at tillering, N6 = All urea at booting, N7 = All N at tillering, N8 = All N at booting, N9 = ½ N at planting + ½ N at tillering, N10 = ½ N 50% emergence + ½ N at tillering, N11 = 1/3rd N at 50% emergence + 1/3rd N at tillering + 1/3rd N at booting, N12 = ½ N at tillering + ½ N at booting.TSW, thousand seed weight (gram); HLW, hectoliter weight. Treatments within a column sharing the same letter are not significantly different.

*significant at the .05 probability level;

**significant at the .01 probability level; ns, non-significant at 0.05 probability level.

development stages recommended for wheat crop, which enhances the maximum exploitation of the genetic potential of the crops.

Results showed that at Adet, the grain yield is ranged from 2.5 t ha$^{-1}$ for applying all N at the booting stage of the crop to 4.89 t ha$^{-1}$ for applying half urea at 50% emergence + half urea at tillering stage of the crop including application of N in DAP at planting (Table 3). At Wonberema, the grain yield ranged from 2.70 t ha$^{-1}$ for applying all N at the booting stage of the crop to 4.63 t ha$^{-1}$ for applying all N at tillering stage of the crop. At Debre Elias, the grain yield ranged from 2.78 t ha$^{-1}$ for applying all N at the booting stage of the crop to 5.23 t ha$^{-1}$ for applying half urea at 50% emergence + half urea at tillering stage of the crop (Table 3). In all locations both biomass and grain yield were significantly highest from applying ½ urea at 50% emergence + ½ urea at tillering including application of N with DAP at planting. Similar to this result, Fresew et al. [48] claimed that the highest grain yield was obtained in three splits of ¼ at sowing, ½ at tillering and ¼ at booting. The other similar result was obtained from the works of Wazir and Akmal [45] who reported that the highest biomass and grain yield and harvest index were recorded when 25%, 25% and 50% of N were applied at sowing, tillering and anthsis of wheat stage. Contrary to this result, Fowler and Brydon [49] and Jan et al. [50] stated that application of fertilizer at sowing only increased wheat grain yield and late fertilization increased grain protein concentration. Efretuei et al. [47] also found that delaying N application until the early stem elongation stage had no detrimental effect on yield. Increased yield with N applied at the earlier and latter growth stages was attributed to a greater number of spikes produced [51] and more tillers per unit area [46]. Efretuei et al. [47] also concluded that

**Table 5. The grain protein content of bread wheat as affected by split and time of nitrogen application[a].**

| N application time | Adet | Wonberema | Debre Elias |
|---|---|---|---|
| **N1** | 16.10[d] | 15.63[c] | 15.13[cd] |
| **N2** | 17.10[abc] | 17.4[a] | 16.25[a] |
| **N3** | 16.48[cd] | 16.80[abc] | 15.72[abc] |
| **N4** | 16.44[cd] | 17.22[ab] | 15.83[ab] |
| **N5** | 16.62[bcd] | 16.22[abc] | 15.38[bcd] |
| **N6** | 17.27[ab] | 17.18[ab] | 16.03[ab] |
| **N7** | 16.55[bcd] | 17.25[ab] | 15.37[bcd] |
| **N8** | 17.78[a] | 16.92[abc] | 15.63[abc] |
| **N9** | 16.18[d] | 15.74[c] | 14.93[d] |
| **N10** | 16.58[bcd] | 15.88[bc] | 14.95[d] |
| **N11** | 16.95[bc] | 17.22[ab] | 15.63[abc] |
| **N12** | 16.98[bc] | 16.48[abc] | 15.85[ab] |
| **Significance level** | ** | ** | ** |
| **CV (%)** | 2.52 | 4.67 | 2.55 |
| **Year** | ** | ** | ** |
| **Year*Treatment** | ** | ** | ** |

[a]Data were combined over years and sites in each experimental locations.N1 = ½ urea at 50% emergence + ½ urea at tillering, N2 = ½ urea at tillering + ½ urea at booting, N3 = 1/3[rd] urea at 50% emergence + 1/3[rd] urea at tillering + 1/3[rd] urea at booting, N4 = 2/3[rd] urea at tillering + 1/3rd urea at booting, N5 = All urea at tillering, N6 = All urea at booting, N7 = All N at tillering, N8 = All N at booting, N9 = ½ N at planting + ½ N at tillering, N10 = ½ N 50% emergence + ½ N at tillering, N11 = 1/3rd N at 50% emergence + 1/3[rd] N at tillering + 1/3[rd] N at booting, N12 = ½ N at tillering + ½ N at booting. Treatments within a column sharing the same letter are not significantly different.

*significant at the 0.05 probability level;

**significant at the 0.01 probability level; ns, non-significant at 0.05 probability level.

the increased recovery of fertilizer N with split application was due to increased crop demand, which was linked to greater rates of leaf area production. However, at Wonberema, the highest and statistically similar effect on grain yield were obtained from the application of ½ urea at 50% emergence + ½ urea at tillering, ½ urea at tillering + ½ urea at booting, 1/3[rd] urea at 50% emergence+1/3[rd] urea at tillering + 1/3[rd] urea at booting, 2/3[rd] urea at tillering + 1/3[rd] urea at booting, all urea at tillering and ½ N at planting + ½ N at tillering (Table 3). Singh et al. [46] reported that nitrogen application in three splits (at sowing, crown root initiation stage and jointing in 1:2:1 ratio) produce higher grain and straw yield than the other splits is in partial conformity with the present study. Split-application of N in three or four times yielded more grain than just two splits, apparently due to high nitrogen fertilizer use efficiency which increases the leaf area and it again increases the photosynthesis rate/photoassimilate that increases the weight per spike. Generally, the grain yield at Adet, Wonberema and Debre Elies was increased by 31%, 14% and 18%, respectively; when N was applied with DAP at planting over without applying N at planting (Table 3). Applying all urea at tillering, all urea at booting, all N at tillering, and all N at booting stage of the crop gave significantly lower grain yield than a split application of N (Fig 2). Efretuei et al. [47] showed that without applying N at sowing and delaying the application of fertilizer N until tillering stage caused a significant reduction in wheat yield. A more likely reason for this could be a reduction in current photosynthesis as a result of reduced leaf area where N application was delayed. Efretuei et al. [47] found that delaying N application until anthesis, compared with applying N just prior to stem extension, caused a reduction in leaf area and therefore reduced assimilation capacity.

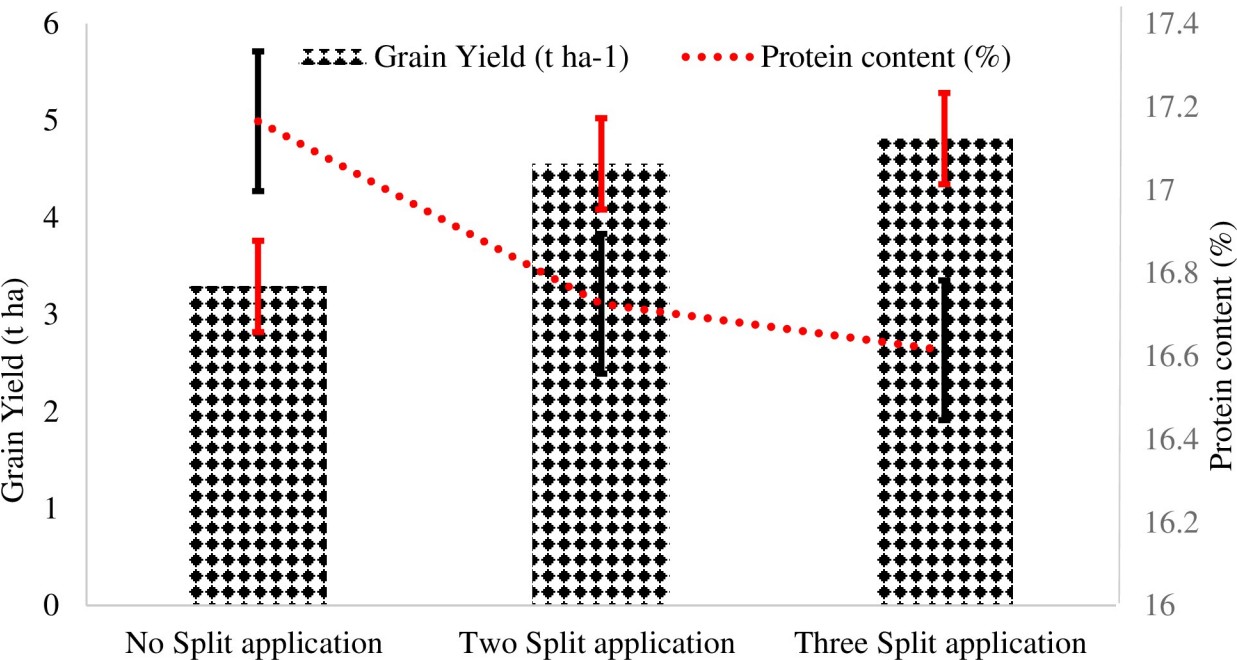

**Fig 2. Effect of frequency of nitrogen application across the crop growth stage on wheat grain yield and protein content.**

## Quality related parameter of bread wheat

The results disclosed that thousand seed weight was not significantly influenced by N split and time of application in all locations (Table 4). In contrary, the grain weight increased when the chemical fertilizers were applied in splits in different crop growth stages as compared to fertilizers application as a single dose at sowing time [46, 50]. Studies revealed intensified wheat fertilization with N resulted in better milling and baking quality through increased hectoliter weight and grain protein content [52]. This is consistent to the present study which showed a significant variation of hectoliter weight and grain protein content between the test sites and the treatments (Tables 4 and 5). At Adet and Wonberema hectoliter weight was significantly affected by N split and time of application while like thousand weight it did not significantly affected by the treatments at Debre Elies. Hectoliter weight is higher at Debere Elias and lower at Adet (Table 4). The addition of N to wheat during the latter growth stages has been observed to increase grain protein but not necessarily yield or test weight [53]. At Adet, hectoliter weight is ranged from 69.73 kg hl$^{-1}$ for applying all N at tillering stage of the crop to 74.23 kg hl$^{-1}$ for applying half urea at 50% emergence + half urea at tillering stage of the crop. At Wonberema, hectoliter weight is ranged from 71.07 kg hl$^{-1}$ for applying 2/3$^{rd}$ urea at tillering +1/3$^{rd}$ urea at booting stage of the crop to 75.57 kg hl$^{-1}$ for applying half urea at planting + half urea at tillering stage of the crop (Table 4). At Adet and Wonberema, the highest hectoliter weight was obtained from the application of ½ N 50% emergence + ½ N at tillering; and ½ N at planting + ½ N at tillering, respectively. In the same respective locations, the lowest hectoliter weight was recorded from the application of all N at tillering; and 2/3$^{rd}$ urea at tillering + 1/3$^{rd}$ urea at booting stages, respectively. However, the latter N treatment is statistically similar effect on hectoliter weight with applying all urea at tillering stage (Table 4). These finding are inconsistent with the findings of

Gobi *et al.* [54] who stated that application of N in three splits at different wheat growth stages plus application of N at sowing gave higher hectoliter weight. The hectoliter weight has also been positively correlated with grain yield [55] (Table 6).

In all locations N split and time of application significantly affected the grain protein content of wheat (Table 5). At Wonberema, grain protein content under different N time of applications ranged from 15.63% for the treatment received half urea at 50% emergence + half urea at tillering to 17.4% for the treatment received half urea at tillering + half urea at booting (Table 5). At Debre Elias grain protein content is ranged from 14.93% for applying half N at 50% emergence + half N at tillering to 16.03% for applying all urea at booting (Table 5). The protein content of wheat grains may vary between 10% - 18% of the total dry matter [56]. The variability in grain protein content due to the environment has been previously described by other researchers and may be explained by temperature and precipitation changes [18, 57, 58]. This relationship between climate variables and grain protein content is very debated and uncertain. The highest grain protein content at Wonberema and Debre Elies was obtained when ½ urea was applied at each wheat tillerng and booting stages. In conformity of this result Ortiz et al. [26] and Ortiz-Monasterio *et al.* [59] reported that split application of N improved grain N content and kernel size. However, at Adet the highest grain protein content was observed when all N was applied at booting stage (with no application of N at planting) (Table 5). This result is parallel to the works of Efretuei *et al.* [47] who stated that delaying the first application of N resulted in an increase in grain N concentration. The absence of a grain protein content reduction as N application was delayed would indicate that either there was sufficient N in the soil to support the plant's needs during this period or that the plant was able to overcome any temporary deficiency in N that occurred during this period. At Adet and Wonberema, the lowest grain protein content and statistically similar effect was observed when ½ urea was applied at each wheat emergence and tillering stages (application of N with DAP at planting) and when ½ N was applied at each wheat planting and tillering stages. At Debre Elies, the lowest grain protein content and statistically similar effect was obtained when ½ N was applied at each wheat planting and tillering stage; and when ½ N was applied at each wheat emergence and tillering stages. At Wonberema and Debre Elies, applying N at planting increased the grain protein content by 0.96% and 2.14%, respectively, over without applying N at planting, while the reverse is true at Adet (Table 5). The presence of a grain protein content increment at the former locations as N was applied at planting would indicate that either there was insufficient N in the soil to support the plant's needs during this period or that the plant was unable to overcome any temporary deficiency in N that occurred during this period.

**Table 6. Pearson's correlation coefficient of agronomic traits with grain yield in each location[a].**

| Characters | Adet | | Wonberema | | Debre Elias | |
|---|---|---|---|---|---|---|
| | r | Sig.level | r | Sig.level | r | Sig.level |
| **Plant height** | 0.67 | < .0001 | 0.81 | < .0001 | 0.63 | < .0001 |
| **Tiller number** | 0.78 | < .0001 | 0.68 | < .0001 | 0.50 | < .0001 |
| **Thousand seed weight** | 0.66 | < .0001 | 0.46 | < .0001 | 0.15 | < .0001 |
| **Above biomass yield** | 0.93 | < .0001 | 0.91 | < .0001 | 0.88 | < .0001 |
| **Hectoliter weight** | 0.66 | < .0001 | 0.58 | < .0001 | 0.48 | < .0001 |
| **Protein content** | -0.78 | 0.0025 | -0.41 | 0.1873 | -0.38 | 0.2189 |

[a]Data were combined across locations and years. r, Pearson's correlation coefficient; Sig.level, significance level.

## Correlation between grain yield and other agronomic and quality parameters of wheat

The correlation analysis between grain yield and the other agronomic and quality parameters of wheat as affected by split and time of N application showed that, at Adet and Wonberema grain yield was highly significantly and strongly correlated in a positive sense with all agronomic attributes (r = 0.46–0.93**) and hectoliter weight (r = 0.58–0.66**) (Table 6). However, at Debre Elies grain yield was highly significantly and positively correlated intermediately with tiller m$^{-2}$ (r = 0.50**) and hectoliter weight (r = 0.48**) and weakly with thousand seed weight (r = 0.15**). Regardless of the strength of r value, grain yield was highly significantly correlated in a negative sense with grain protein content (r = -0.78–0.38**) (Table 6). It appeared that when yields were low, sufficient N was generally available at the time of grain filling to lead to an increase in grain protein content. This observation is similar to that of Alcoz *et al.* [51]. Previous studies have also reported that negative relationship between grain protein content and grain yield [18, 60]. Similar to other findings, simple linear regression analysis in the present study indicated that grain yield and protein content are inversely affected as the frequency of N application increased across wheat growing stages (Fig 3). Increasing N availability generally increases yield more than protein until a yield maximum is reached, whereupon protein levels increase if N is increased further [61, 62].

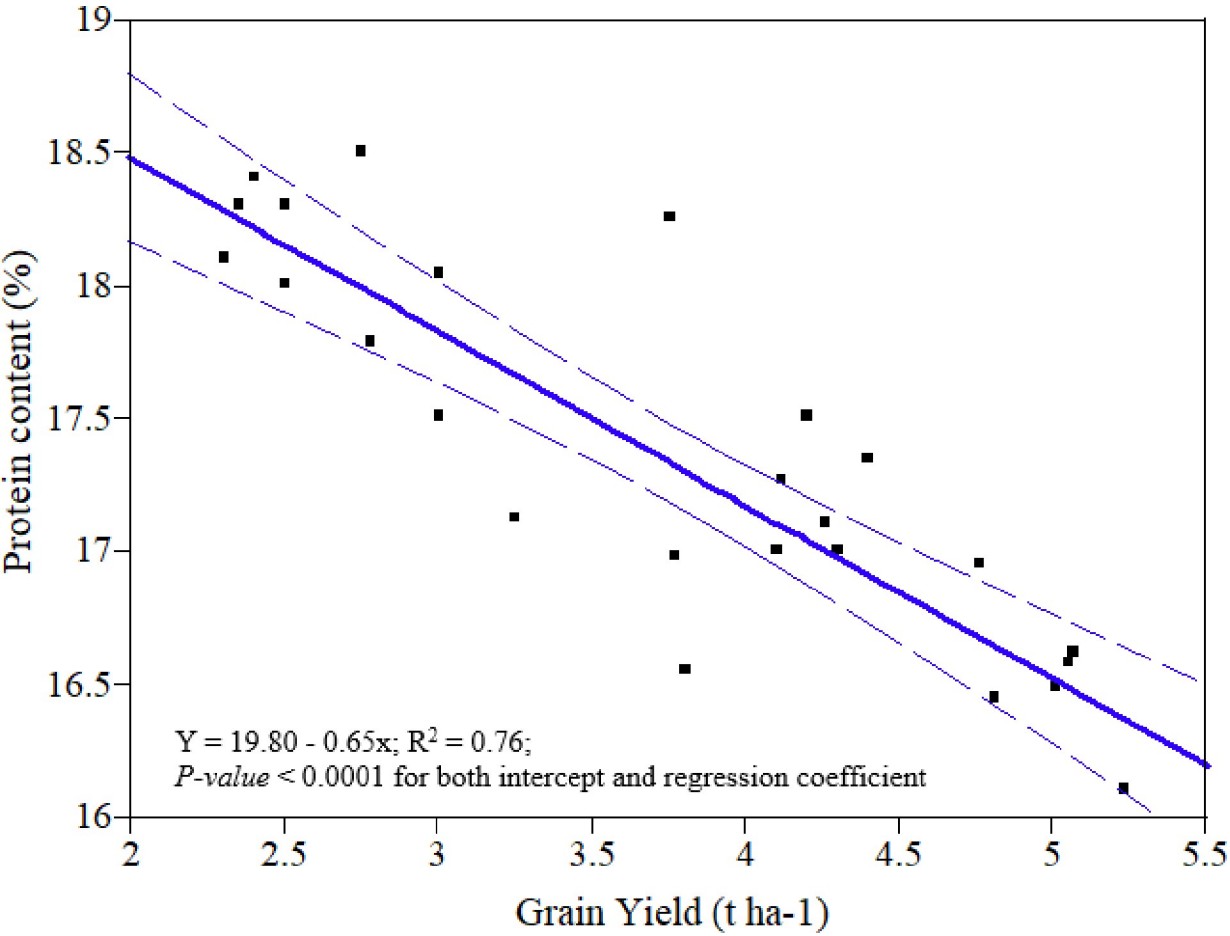

**Fig 3. Simple linear regression analysis for grain yield and protein content of bread wheat as affected by nitrogen split and time of application.**

## Conclusions

The study showed wheat grain yield and protein content was highly influenced by the environment and indirectly correlated with each other as affected by N time of applications. The grain yield at Adet, Wonberema and Debre Elies was increased by 31%, 14% and 18%, respectively when N was applied with DAP at sowing over the blanket recommendations. At all locations, grain protein content decreased as the split application increased. Thus, depending on the purpose of the producers, it can concluded that application of ½ urea at 50% emergence + ½ urea at tillering with the application of N with DAP at sowing gave maximum wheat grain yield, while optimum grain protein content was obtained when N was applied after the crop is emerged and would be used in most dominant wheat producing areas of northwestern Ethiopia. Further study should be conducted on split application of Blended fertilizers (NPS, NPSBZN etc.)

## Supporting information

**S1 Data.**
(XLSX)

## Author Contributions

**Conceptualization:** Bitwoded Derebe, Yayeh Bitew, Fikeremariam Asargew, Gobezie Chakelie.

**Data curation:** Bitwoded Derebe, Yayeh Bitew.

**Formal analysis:** Bitwoded Derebe, Yayeh Bitew.

**Investigation:** Bitwoded Derebe, Yayeh Bitew, Fikeremariam Asargew, Gobezie Chakelie.

**Methodology:** Bitwoded Derebe, Yayeh Bitew, Fikeremariam Asargew, Gobezie Chakelie.

**Resources:** Bitwoded Derebe.

**Software:** Bitwoded Derebe, Yayeh Bitew.

**Supervision:** Bitwoded Derebe, Yayeh Bitew, Fikeremariam Asargew, Gobezie Chakelie.

**Validation:** Bitwoded Derebe, Yayeh Bitew, Fikeremariam Asargew, Gobezie Chakelie.

**Visualization:** Bitwoded Derebe, Yayeh Bitew, Fikeremariam Asargew, Gobezie Chakelie.

**Writing – original draft:** Bitwoded Derebe, Yayeh Bitew.

**Writing – review & editing:** Bitwoded Derebe, Yayeh Bitew, Fikeremariam Asargew, Gobezie Chakelie.

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
