## [Decision Letter · Decision Letter 0]

30 Aug 2022

PONE-D-22-18317Optimizing Nitrogen Fertilizer Split and Time of Application to Bread Wheat (Triticum Aestivum L.) Grain Yield and Quality in Northwestern EthiopiaPLOS ONE

Dear Dr. Bitew,

Thank you for submitting your manuscript to PLOS ONE. After careful consideration, we feel that it has merit but does not fully meet PLOS ONE’s publication criteria as it currently stands. Therefore, we invite you to submit a revised version of the manuscript that addresses the points raised during the review process.

We look forward to receiving your revised manuscript.

Kind regards,

Cataldo Pulvento

Academic Editor

PLOS ONE

Journal Requirements:

3. We note that Figure 1 in your submission contain map image which may be copyrighted. All PLOS content is published under the Creative Commons Attribution License (CC BY 4.0), which means that the manuscript, images, and Supporting Information files will be freely available online, and any third party is permitted to access, download, copy, distribute, and use these materials in any way, even commercially, with proper attribution. For these reasons, we cannot publish previously copyrighted maps or satellite images created using proprietary data, such as Google software (Google Maps, Street View, and Earth). For more information, see our copyright guidelines: http://journals.plos.org/plosone/s/licenses-and-copyright.

Additional Editor Comments:

Dear Authors

 Please revise the manuscript according to the reviewers comments.

Reviewers' comments:

Reviewer's Responses to Questions

**Comments to the Author**

1. Is the manuscript technically sound, and do the data support the conclusions?

Reviewer #1: Yes

Reviewer #2: Yes

2. Has the statistical analysis been performed appropriately and rigorously? 

Reviewer #1: Yes

Reviewer #2: Yes

3. Have the authors made all data underlying the findings in their manuscript fully available?

Reviewer #1: Yes

Reviewer #2: Yes

4. Is the manuscript presented in an intelligible fashion and written in standard English?

Reviewer #1: No

Reviewer #2: Yes

5. Review Comments to the Author

Reviewer #1: Abstract

It’s missing full stop and the end of the first sentence.

Name the treatments

What is the difference between all urea at booting and all N at booting

it can concluded, correct sentence is “It can be concluded”.

The abstract is sufficiently clear and focused on the main results of the work

Introduction

“Despite its importance, the productivity of wheat in Ethiopia under rain fed season is low

particularly in Amhara region (2.79)” what is the unit of 2.79?

“Low soil fertility, improper fertilizer application, lack of appropriate seeding, fertilizer

rate; and planting methods and pests are among the major constraints limiting wheat production

in Ethiopia [12, 13].”

Write fertilizer rate instead of fertilizer rate, replace “;” by “,” and remove “and” before planting methods

“To reduce these

challenges increasing the N use efficiency through different strategies such as (i) use of

appropriate type of fertilizer and application time during the crop growing period, (ii) application

of optimum N fertilizer rate, (iii) appropriate method of application, (iv) use of appropriate

cultivar and (v) application at the appropriate climatic conditions [21].”

There is no verb in this sentence

“Thus, nitrogen scheduling plays a vital role in

the growth, production, and quality of wheat as well as in its use efficiency [20].”

Some articles focused on the effect of N splitting on wheat https://doi.org/10.1016/j.eja.2004.12.001;

https://doi.org/10.1016/S0378-4290(03)00167-9;

https://doi.org/ 10.1016/S0378-4290(03)00167-9

https://doi.org/ 10.3389/fpls.2016.00738

“Thus, the main objective of this research was to determine the

appropriate N fertilizer application time for maximum bread wheat yield production with

optimum protein contenet.” Typing error prtein contenet. Write protein content…

“…wheat growing stags with various ratios of N across…” write growing stages

Materials and Methods

Description of Experimental Sites

A field experiment was conducted in three wheat production potential areas of…

Potential areas of…

The manuscript is full of typing errors

“… districts of Amhara Region, Ethiopia (Figure 1) …” Follow the journal's instructions for citing figures and tables in the manuscript.

“Generally, the rainfall in the study areas follows a dominantly unimodal distribution with the

main peak in June to September,” Does this mean June is rainy as July, August and September? Main peak should be the rainiest month

“Geographically, Adet is located at 11o16' N latitude and 37o29' E longitude with an altitude of 2240 meters above sea level (m.a.s.l.) It has a long year mean annual rainfall of 1211mm with a minimum and maximum temperature of 11.57°C and 26.89°C respectively [33]. Wonberema is located at 10°27' N latitude and 37°56' E, longitude with an altitude of 2600 m.a.s.l. It has a long year mean annual rainfall of 1211mm with a minimum and maximum temperature of 17°C and 25°C, respectively”

The mean annual rainfall mentioned here is less than the rainfall recorded during the growing season in each of the 3 areas (Figure 2). How do authors explain this? Are they mean annual rainfall calculated over a period?

“In Debre Elias and Wonberema, the rain falls increased from 2014 and 2015 by

71.43% and 14.29%, respectively, ...” write ...from 2014 to 2015 by 71.43% and 14.29%, respectively,

“At Adet the soil textural class is clay with the soil pH, OC (g kg-1), TN (mg kg-1), Ava.P (mg kg-1), CEC (meq100g-1) of 5.4, 10.7, 966, 27.69 and 31.2, respectively [35]. The soil pH , OC(%), TN (%), Ava.P (ppm) and Av.K (ppm) at Wonberema were 5.7, 3.1, 0.3, 369 and 127, respectively with sandy loam textural lass [36]. The soil pH, OM (%), TN (%), Ava.P (mg kg-1) and CEC (Cmole(+) kg-1) at Debre Elias were 5.12, 2.63, 0.14, 2.53 and 25.02, respectively”

Better to present this in a table and make comments.

Having 966 mgN/kg of soil at Adet, no need to conduct such study on N splitting there. The soil contents much N than plant required. Chek this result. Same remark for Ava.P at Wonderema (369mg/kg). Why authors present TN results in different unit (mg kg-1 and %) in the same study. Please harmonize it. ppm = mg kg-1

Morever, the soil characterics presented are different from location to location. For instance, at Wonberema, authors don’t present the soil texture and CEC value and at Debre no soil texture presented. Harmonize this since soil texture and CEC impact plant response

Treatments, Experiment Design and Planting Proceduers

1. ½ urea at 50% emergence (Zadoks Growth Stage 11-18) + ½ urea at tillering (Zadoks

Growth Stage 21-26)

2. ½ urea at tillering (Zadoks Growth Stage 21-26) + ½ urea at booting (Zadoks Growth

stage 41- 45)

8

3. 1/3rd urea at 50% emergence (Zadoks Growth Stage 11-18) + 1/3rd urea at tillering

(Zadoks Growth Stage 21-26) + 1/3rd urea at booting (Zadoks Growth stage 41- 45)

4. 2/3rd urea at tillering (Zadoks Growth Stage 21-26) + 1/3rd urea at booting (Zadoks

Growth stage 41-45)

5. All urea at tillering (Zadoks Growth Stage 21- 26)

6. All urea at booting (Zadoks Growth stage 41- 45)

7. All N at tillering (Zadoks Growth Stage 21- 26)

8. All N at booting (Zadoks Growth stage 41- 45)

9. ½ N at planting (Zadoks Growth Stage 00) + ½ N at tillering (Zadoks Growth Stage 21-

26)

10. ½ N 50% emergence + ½ N at tillering (Zadoks Growth Stage 21- 26)

11. 1/3rd N at 50% emergence (Zadoks Growth Stage 11-18) + 1/3rd N at tillering (Zadoks

Growth Stage 21-26) + 1/3rd N at booting (Zadoks Growth stage 41- 45)

12. ½ N at tillering (Zadoks Growth Stage 21-26) + ½ N at booting (Zadoks Growth stage

41- 45)

I suggest authors to present this in table to facilitate the reading to readers

For instance, a table like the following

Treatment Nitrogen application

1st application 2nd application 3rd application

1

2

“Note that, from treatment 1 to 6, N was applied with Di-Ammonium Phosphate (DAP) at planting and from treatment 7 to 12 no N was applied at planting instead TSP (Tri- Supper Phosphate) was used.”

“Note that, from treatment 1 to 6, N was applied with Di-Ammonium Phosphate (DAP) at

planting and from treatment 7 to 12 no N was applied at planting instead TSP (Tri- Supper

Phosphate) was used.”

What about TSP application for treatment 1 to 6? As TSP was applied to all plot, I suggest authors to state “Note that, from treatment 1 to 6, N was applied with Di-Ammonium Phosphate (DAP) at planting and from treatment 7 to 12 no N was applied at planting.” Continue with the rest of sentence seems like no P was applied for treatment 1 to 6 and this change completely the objective of the study. But few lignes after, we can read

“…DAP and TSP were the sources of P2O5 and DAP and urea and DAP were the sources of N…”

There is one more DAP in the sentence.

There is one more “DAP” in the sentence. Rephrase it as :

“…DAP and TSP were the sources of P2O5 and DAP and urea were the sources of N…”

 , write adjusted yield

“At crop maturity, a subsample from each net plot was harvested at ground level, oven-dried at 70°C until constant weight was reached for dry weight determination and partitioned into straw and grain.”

Why this yield for protein content was not adjusted to 12.5% the moisture?

Results and Discussion

Effect of locations and growing seasons

From Table 1 to 4, all significance shown is P < 0.01, not P < 0.001

Under Table 3 “**significant at the .01 probability level” one asterisk is smaller than the second. Harmonise

Yield and yield component of bread wheat

“Results showed in all locations N split and application time was significantly (P<0.01) …” Results shown…”

“Results showed in all locations N split and application time was significantly (P<0.01)

affected plant height, total tillers m-2, biomass and grain yield (Table 1).” Add Table 2 also in bracket. Grain and biomass yields are presented in Table 2

“… The possible explanation of the positive effect of split N application compared to single N application at later growth stages is gaseous losses from applied topdressing. …”

This is mainly due to the poor response of plants at advanced growth stages that cause N leaching or volatilization

“Specifically, the highest tiller m-2 at Adet and Debre Elies was obtained from the application of ½ urea at 50% emergence + ½ urea at tillering plus adition of N with DAP at planting, …” Addition instead of adition

“… production of tillers and ears that result in grains [18), …” The citation should be in bracket

Sometimes authors write Wonberema, sometimes Wonberima in tables or Woneberema, what is the right handwritting

Figure 3: The standard deviation/error bars for the protein content are not very visible for 2 and 3 split applications. Change the color. Either put it in red like the curve

“The non-responsiveness of hectoliter weight to N application at Debre Elies was in line with

that of Dawit et al. [53] who reported a non-significant effect of N rates on hectoliter weight of

bread wheat.”

The reference [53] refers to maize not bread. Moreover, the current study focus on the effect of N split and time of application not N rates. So the comparison is not appropriate.

“In all locations N split and time of application significantly affected the grain protein content of wheat (Table 4).”

This sentence should be moved to the beginning of the next section on grain protein content

“[55] ( Iqbal et al., 2012).” Delete the citation in parenthesis

“The highest grain protein content at Woneberema and Debre Elies was absorbed when…” Absorved instead of Obserbed

“… when all N was applied at booting stage (with no application of N at planting) Table 4). This result is parallel to th works of Efretuei et al. [47]” Typing errors in the sentence (parenthesis and “e” miss)

“The absence of a grain protein content reduction as N application was delayed would indicate that either there was sufficient N in the soil to support the plant’s needs during this period or that the plant was able to overcome any temporary deficiency in N that occurred during this period.

Authors (Smith et al., 1991; Mi et al., 2000; Woolfolk et al., 2002), showed that delaying N application till booting, anthesis or flowering increase mainly grain protein content than grain yield

Correlation between grain yield and other agronomic and quality

parameters of wheat

There are no table 5 and 6 in the manuscript as announced in this section

In table 7, the pearson correlation coefficient is represented by both "r" and "R. I suggest that the authors be consistent and rigorous in writing

References

Some references are incomplete. Names of editions and cities/countries for quoted books are missing; numbers and pages of some quoted journals are also missing. Some examples

[1]. FAO (Food and Agriculture Organization). Crop Prospects and Food Situation. 2014; Pp5-7

It is missing the edition and place of publication in this reference

[5]. CSA (Central Statistical Agency) 2018. Agricultural Sample Survey, report on area and

production of major crops. Statistical bulletin……2018 ;. Addis Ababa, Ethiopia

Numero ??? Page?

[14]. Amsal T and Tanner D. Effects of Fertilizer application on N and P uptake, recovery and

use efficiency of bread wheat grown on two soil types in central Ethiopia. Ethiopian

Journal of Natural Resources.2001;

Complete with the number and pages

MY CONCLUSION ON THE MANUSCRIPT

Overall, it is a good scientific contribution and provides a solution on N management to wheat farmers especially in Ethiopia. The introduction and materials and methods are sufficiently documented and the conclusion support the study findings. However, the authors lacked rigor in the manuscript writing : many typing errors, grammar and phrasing issues , the journal's instructions to the authors not followed for the citation of figures, the announced tables not included, incomplete and sometimes inappropriate references.

The manuscript can be accepted after correction of identified deficiencies.

Reviewer #2: The manuscript is interesting and written in good way with the eception of the following minor comments

Grammer and typological error

Consistency

Most of the Suppprting ideas are out of the topic

References should be rerite again correctly by including year, volume, issue and page and publisher and place of publisher for journal and books used as asources

6. PLOS authors have the option to publish the peer review history of their article (what does this mean?). If published, this will include your full peer review and any attached files.

Reviewer #1: No

Reviewer #2: **Yes: **Fenta Assefa Bogale

---

## [Author Response · Author response to Decision Letter 0]

5 Sep 2022

Reviewer comment Author response

Reviewer 1

Optimizing Nitrogen Fertilizer Split and Time of Application to Bread Wheat (Triticum Aestivum L.) Grain Yield and Quality in Northwestern Ethiopia

PLOS ONE Optimizing Time and Split Application of Nitrogen Fertilizer to harness Grain Yield and Quality of Bread Wheat (Triticum Aestivum L.) in Northwestern Ethiopia

Sowing is appropriate word

 Accepted the comment

How you can know, there was no any genotype tria in the experiment The yield and quality are varied across the environments

Consistency, once planting and the other time sowing. Anyways sowing is the appropriate word

 corrected

Is it the amount of nitrogen fertilizer applied in split increased you mean ?? If it is so please add words indicating amount/rates of N application in split form? At all locations, grain protein content decreased as the number of N split application increased one to 3 times.

What about the Economic analaysis?? Just include it as it is very crucial to recommend for farmers based on the cost they incur and benefits gained??

 included

Key words should be in alphabetic Key words: grain protein content; grain yield; hectoliter weight; wheat

is it the right reasoning? Instead it is due to high nitrogen fertilizer use efficiency which increases the canopy/leaf area and it again increases the photosynthesis rate/photoassimilate that increases the grain weight. Corrected 

Reference Corrected 

Reviewer 2

It’s missing full stop and the end of the first sentence. 

Name the treatments Descried I the document

What is the difference between all urea at booting and all N at booting

 Same but to differentiate treatment 1 to 6, (N was applied with Di-Ammonium Phosphate (DAP) at sowing) and treatment 7 to 12 (no N was applied at sowing instead TSP (Tri- Supper Phosphate) was used). Please see in the materials and methods

it can concluded, correct sentence is “It can be concluded”.

 Corrected based on the comments

“Despite its importance, the productivity of wheat in Ethiopia under rain fed season is low

particularly in Amhara region (2.79)” what is the unit of 2.79?

 2.79 tone/ha

“Low soil fertility, improper fertilizer application, lack of appropriate seeding, feretilizer

rate; and planting methods and pests are among the major constraints limiting wheat production

in Ethiopia [12, 13].”

 Low soil fertility, improper fertilizer application, lack of appropriate seeding rate, fertilizer rate, planting methods and pests are among the major constraints limiting wheat production in Ethiopia [12, 13].

To reduce these

challenges increasing the N use efficiency through different strategies such as (i) use of

appropriate type of fertilizer and application time during the crop growing period, (ii) application

of optimum N fertilizer rate, (iii) appropriate method of application, (iv) use of appropriate

cultivar and (v) application at the appropriate climatic conditions [21].”

 To reduce these challenges increasing the N use efficiency through different strategies is necessary. Some of the strategies includes

“Thus, the main objective of this research was to determine the

appropriate N fertilizer application time for maximum bread wheat yield production with

optimum protein contenet.” Typing error prtein contenet. Write protein content…

 Thus, the main objective of this research was to determine the appropriate N fertilizer application time for maximum bread wheat yield production with optimum protein content.

“…wheat growing stags with various ratios of N across…” write growing stages

 corrected

A field experiment was conducted in three wheat production potentail areas of…

Potential areas of…

 A field experiment was conducted in three wheat production potential areas

“… districts of Amhara Region, Ethiopia (Figure 1) …” Follow the journal's instructions for citing figures and tables in the manuscript. 

“Generally, the rainfall in the study areas follows a dominantly unimodal distribution with the

main peak in June to September,” Does this mean June is rainy as July, August and September? Main peak should be the rainiest month

 No sir, highest rainfall has been found in those months

The mean annual rainfall mentioned here is less than the rainfall recorded during the growing season in each of the 3 areas (Figure 2). How do authors explain this? Are they mean annual rainfall calculated over a period?

 Yes sir, most of the time The mean annual rainfall over the period (30 years) is less than the rainfall recorded during the growing season in each of the 3 areas.

“In Debre Elias and Wonberema, the rain falls increased from 2014 and 2015 by

71.43% and 14.29%, respectively, ...” write ...from 2014 to 2015 by 71.43% and 14.29%, respectively, 

 In Debre Elias and Wonberema, the rain fall increased from 2014 to 2015 by 71.43% and 14.29%, respectively, while at Adet it was relatively constant across years

At Adet the soil textural class is clay with the soil pH, OC (g kg-1), TN (mg kg-1), Ava.P (mg kg-1), CEC (meq100g-1) of 5.4, 10.7, 966, 27.69 and 31.2, respectively [35]. The soil pH , OC(%), TN (%), Ava.P (ppm) and Av.K (ppm) at Wonberema were 5.7, 3.1, 0.3, 369 and 127, respectively with sandy loam textural lass [36]. The soil pH, OM (%), TN (%), Ava.P (mg kg-1) and CEC (Cmole(+) kg-1) at Debre Elias were 5.12, 2.63, 0.14, 2.53 and 25.02, respectively”

Better to present this in a table and make comments.

Having 966 mgN/kg of soil at Adet, no need to conduct such study on N splitting there. The soil contents much N than plant required. Chek this result. Same remark for Ava.P at Wonderema (369mg/kg). Why authors present TN results in different unit (mg kg-1 and %) in the same study. Please harmonize it. ppm = mg kg-

Morever, the soil characterics presented are different from location to location. For instance, at Wonberema, authors don’t present the soil texture and CEC value and at Debre no soil texture presented. Harmonize this since soil texture and CEC impact plant response

 At Adet the soil textural class is clay [35] while at Wonberema [36] and Debre Elias [37] the soil textural class is sandy loam. The experimental site at Adet constitutes pH 5.4, organic carbon 2.47% Total nitrogen 0.8%, available phosphorus 1.98 ppm and cation exchange capacity 31.2 Cmole(+) kg-1 [35] while the respective soil nutrients at Debre Elias were 5.12, 2.63%, 0.14% , 2.53pp and 25.02 Cmole(+) kg-1 [37]. At Wonberema, pH, organic carbon, total nitrogen and available phosphorus were 572, 3.1%, O.3%, 3.69ppm, respectively [36].

I suggest authors to present this in table to facilitate the reading to readers

For instance, a table like the following

Treatment Nitrogen application

 1st application 2nd application 3rd application

1 

2 

Note that, from treatment 1 to 6, P was applied in the form of Di-Ammonium Phosphate (DAP) while from treatment 7 to 12, P was applied in the form of TSP (Tri- Supper Phosphate) at sowing. Thus, N was applied at sowing only in the former treatments (1 to 6)

“Note that, from treatment 1 to 6, N wa++

s applied with Di-Ammonium Phosphate (DAP) at planting and from treatment 7 to 12 no N was applied at planting instead TSP (Tri- Supper Phosphate) was used.”

“Note that, from treatment 1 to 6, N was applied with Di-Ammonium Phosphate (DAP) at

planting and from treatment 7 to 12 no N was applied at planting instead TSP (Tri- Supper

Phosphate) was used.” 

What about TSP application for treatment 1 to 6? As TSP was applied to all plot, I suggest authors to state “Note that, from treatment 1 to 6, N was applied with Di-Ammonium Phosphate (DAP) at planting and from treatment 7 to 12 no N was applied at planting.” Continue with the rest of sentence seems like no P was applied for treatment 1 to 6 and this change completely the objective of the study. But few lignes after, we can read 

“…DAP and TSP were the sources of P2O5 and DAP and urea and DAP were the sources of N…”

There is one more DAP in the sentence. 

There is one more “DAP” in the sentence. Rephrase it as :

“…DAP and TSP were the sources of P2O5 and DAP and urea were the sources of N…”

 Table 2: Description of the experimental treatments

TrtNo. Nitrogen application 

 1st application 2nd application 3rd application

1 1/3rd urea at 50% emergence ½ urea at tillering -

2 ½ urea at tillering ½ urea at booting -

3 1/3rd urea at 50% emergence 1/3rd urea at tillering 1/3rd urea at booting 

4 2/3rd urea at tillering 1/3rd urea at booting -

5 All urea at tillering - -

6 All urea at booting - -

7 All N at tillering - -

8 All N at booting - -

9 ½ N at sowing ½ N at tillering -

10 ½ N 50% emergence ½ N at tillering -

11 1/3rd N at 50% 1/3rd N at tillering 1/3rd N at booting 

12 ½ N at tillering ½ N at booting -

TrtNo.�Treatment number. Sowing Zadoks Growth Stage 00; emergence Zadoks Growth Stage 11-18; tillering�-Zadoks Growth Stage 21-26); booting Zadoks Growth stage 41- 45.

Note that, Di-Ammonium Phosphate (DAP) and TSP (Tri- Supper Phosphate) were the sources of P2O5 and DAP and UREA were the sources of N. However, at sowing P2O5 was applied in the form of DAP from treatment 1 to 6, while from treatment 7 to 12, P2O5 was applied in the form of TSP.

“At crop maturity, a subsample from each net plot was harvested at ground level, oven-dried at 70°C until constant weight was reached for dry weight determination and partitioned into straw and grain.”

Why this yield for protein content was not adjusted to 12.5% the moisture?

 From the above dried, cleaned and adjusted yield a subsample from each net plot was taken and milled. Then grain N content of the samples were determined using the micro-Kjeldahl method as stated by American Association of Cereal Chemists (AACC) [41]. The protein content of wheat flour samples for each treatment was performed by following AACC [42] using a 5.7 conversion factor as follows and expressed in dry matter %.

From Table 1 to 4, all significance shown is P < 0.01, not P < 0.001

Under Table 3 “**significant at the .01 probability level” one asterisk is smaller than the second. Harmonise

 Corrected based on the comments

“Results showed in all locations N split and application time was significantly (P<0.01) …” Results shown…”

“Results showed in all locations N split and application time was significantly (P<0.01)

affected plant height, total tillers m-2, biomass and grain yield (Table 1).” Add Table 2 also in bracket. Grain and biomass yields are presented in Table 2

“… The possible explanation of the positive effect of split N application compared to single N application at later growth stages is gaseous losses from applied topdressing. …” 

 Results shown in all locations N split and application time was significantly (P<0.01) affected plant height, total tillers m-2, biomass and grain yield (Table 3 and 4).

The possible explanation of the positive effect of split N application compared to single N application at later growth stages is gaseous losses from applied topdressing

This is mainly due to the poor response of plants at advanced growth stages that cause N leaching or volatilization 

“Specifically, the highest tiller m-2 at Adet and Debre Elies was obtained from the application of ½ urea at 50% emergence + ½ urea at tillering plus adition of N with DAP at planting, …” Addition instead of adition

 This is mainly due to the poor response of plants at advanced growth stages that cause N leaching or volatilization. Specifically, the highest tiller m-2 at Adet and Debre Elies was obtained from the application of ½ urea at 50% emergence + ½ urea at tillering plus addition of N with DAP at planting,

“… production of tillers and ears that result in grains [18), …” The citation should be in bracket

Sometimes authors write Wonberema, sometimes Wonberima in tables or Woneberema, what is the right handwritting

 [18]

Wonberema

Figure 3: The standard deviation/error bars for the protein content are not very visible for 2 and 3 split applications. Change the color. Either put it in red like the curve

 Corrected 

“The non-responsiveness of hectoliter weight to N application at Debre Elies was in line with

that of Dawit et al. [53] who reported a non-significant effect of N rates on hectoliter weight of

bread wheat.” 

 Corrected 

“In all locations N split and time of application significantly affected the grain protein content of wheat (Table 4).”

This sentence should be moved to the beginning of the next section on grain protein content

 Corrected 

“[55] ( Iqbal et al., 2012).” Delete the citation in parenthesis

“The highest grain protein content at Woneberema and Debre Elies was absorbed when…” Absorved instead of Obserbed

 Corrected

The highest grain protein content at Wonberema and Debre Elies was obtained when ½ urea was applied at each wheat tillerng and booting stages

Correlation between grain yield and other agronomic and quality

parameters of wheat

There are no table 5 and 6 in the manuscript as announced in this section

In table 7, the pearson correlation coefficient is represented by both "r" and "R. I suggest that the authors be consistent and rigorous in writing

 Corrected based on the comments

References

Some references are incomplete. Names of editions and cities/countries for quoted books are missing; numbers and pages of some quoted journals are also missing. Some examples

 [1]. FAO (Food and Agriculture Organization). Crop Prospects and Food Situation. 2014; Pp5-7

It is missing the edition and place of publication in this reference

[5]. CSA (Central Statistical Agency) 2018. Agricultural Sample Survey, report on area and

production of major crops. Statistical bulletin……2018 ;. Addis Ababa, Ethiopia

Numero ??? Page?

[14]. Amsal T and Tanner D. Effects of Fertilizer application on N and P uptake, recovery and

use efficiency of bread wheat grown on two soil types in central Ethiopia. Ethiopian

Journal of Natural Resources.2001;

Complete with the number and pages

 All references are Corrected

---

## [Decision Letter · Decision Letter 1]

24 Oct 2022

PONE-D-22-18317R1Optimizing Time and Split Application of Nitrogen Fertilizer to Harness Grain Yield and Quality of Bread Wheat (Triticum Aestivum L.) in Northwestern EthiopiaPLOS ONE

Dear Dr. Bitew,

Thank you for submitting your manuscript to PLOS ONE. After careful consideration, we feel that it has merit but does not fully meet PLOS ONE’s publication criteria as it currently stands. Therefore, we invite you to submit a revised version of the manuscript that addresses the points raised during the review process.Please revise the manuscript according to the reviewer comments

We look forward to receiving your revised manuscript.

Kind regards,

Cataldo Pulvento

Academic Editor

PLOS ONE

Journal Requirements:

Reviewers' comments:

Reviewer's Responses to Questions

**Comments to the Author**

1. If the authors have adequately addressed your comments raised in a previous round of review and you feel that this manuscript is now acceptable for publication, you may indicate that here to bypass the “Comments to the Author” section, enter your conflict of interest statement in the “Confidential to Editor” section, and submit your "Accept" recommendation.

Reviewer #1: (No Response)

2. Is the manuscript technically sound, and do the data support the conclusions?

Reviewer #1: Yes

3. Has the statistical analysis been performed appropriately and rigorously? 

Reviewer #1: Yes

4. Have the authors made all data underlying the findings in their manuscript fully available?

Reviewer #1: Yes

5. Is the manuscript presented in an intelligible fashion and written in standard English?

Reviewer #1: Yes

6. Review Comments to the Author

Reviewer #1: The authors have taken into account the majority of the recommendations. However, some minor corrections are still needed before publication

7. PLOS authors have the option to publish the peer review history of their article (what does this mean?). If published, this will include your full peer review and any attached files.

Reviewer #1: No

---

## [Author Response · Author response to Decision Letter 1]

1 Nov 2022

Response to reviewer 

The mean annual rainfall mentioned here is less than the rainfall recorded during the growing season in each of the 3 areas (Figure 2). How do authors explain this? Are they mean annual rainfall calculated over a period? Yes sir, most of the time. The mean annual rainfall over the period (30 years) is less than the rainfall recorded during the growing season in each of the 3 areas.

For better understanding indicate the period over which the mean annual rainfall is calculated

Response to Reviewers: It has a long year mean (30 years) annual rainfall of 1211mm with a minimum and maximum temperature of 11.57°C and 26.89°C, respectively

… while the respective soil nutrients at Debre Elias were 5.12, 2.63%, 0.14% , 2.53pp and 25.02 Cmole (+) kg-1 [37]. At Wonberema, pH, organic carbon, total nitrogen and available phosphorus were 572, 3.1%, O.3%, 3.69ppm, respectively [36].

Still have typing errors in this sentence. Correct them

2.53 ppm

pH at Wonberema was 5.72 

Total nitrogen at Wonberema 0.3% not O.3% (the zero written with letter “O”

Response to Reviewers. At Adet the soil textural class is clay [35] while at Wonberema [36] and Debre Elias [37] the soil textural class is sandy loam. The experimental site at Adet constitutes an average pH of 5.4, organic carbon of 2.47%, total nitrogen of 0.8%, available phosphorus of 1.98 ppm and cation exchange capacity of 31.2 Cmole (+) kg-1 [35] while the respective soil nutrients at Debre Elias were 5.12, 2.63%, 0.14%, 2.53ppm and 25.02 Cmole(+) kg-1 [37]. At Wonberema, pH, organic carbon, total nitrogen and available phosphorus were 5.72, 3.1%, 0.3%, 3.69ppm, respectively [36].

Data collection

Ajusted yield instead of adujested yield

Response to Reviewers. Corrected based on the comment

Results and discussion

This is mainly due to the poor response of plants at advanced growth stages that cause N leaching or volatilization. This sentence supports the following statement “The possible explanation of the positive effect of split N application compared to single N application at later growth stages is gaseous losses from applied topdressing. It is not to explain the increase in the number of tillers with split applications. Please remove it 

Response to Reviewers. Corrected based on the comment

In table 6, the pearson correlation coefficient ‘r’ is written in the same table with upper case R and lower case r. Be coherent and rigourous. Either “r” or “R” not both at the same time

Response to Reviewers. Corrected based on the comment

---

## [Editor Report · Decision Letter 2]

2 Dec 2022

Optimizing Time and Split Application of Nitrogen Fertilizer to Harness Grain Yield and Quality of Bread Wheat ( Triticum Aestivum L.) in Northwestern Ethiopia

PONE-D-22-18317R2

Dear Dr. Bitew,

We’re pleased to inform you that your manuscript has been judged scientifically suitable for publication and will be formally accepted for publication once it meets all outstanding technical requirements.

Kind regards,

Cataldo Pulvento

Academic Editor

PLOS ONE
---

## [Editor Report · Acceptance letter]

5 Dec 2022

PONE-D-22-18317R2 

Optimizing Time and Split Application of Nitrogen Fertilizer to Harness Grain Yield and Quality of Bread Wheat (Triticum Aestivum L.) in Northwestern Ethiopia 

Dear Dr. Bitew:

I'm pleased to inform you that your manuscript has been deemed suitable for publication in PLOS ONE. Congratulations! Your manuscript is now with our production department. 

Kind regards, 

on behalf of

Dr. Cataldo Pulvento 

Academic Editor

PLOS ONE